# Case-area targeted preventive interventions to interrupt cholera transmission: Current implementation practices and lessons learned

**Mustafa Sikder**[1,2], **Chiara Altare**[1,2], **Shannon Doocy**[1,2], **Daniella Trowbridge**[1,2], **Gurpreet Kaur**[1,2], **Natasha Kaushal**[1,2], **Emily Lyles**[1,2], **Daniele Lantagne**[3], **Andrew S. Azman**[1,2], **Paul Spiegel**[1,2]*

1 Johns Hopkins Bloomberg School of Public Health, Baltimore, Maryland, United States of America,
2 Johns Hopkins Center for Humanitarian Health, Baltimore, Maryland, United States of America,
3 Consultant, Public Health Engineer, Boston, Massachusetts, United States of America

* pbspiegel@jhu.edu

## Abstract

### Background

Cholera is a major cause of mortality and morbidity in low-resource and humanitarian settings. It is transmitted by fecal-oral route, and the infection risk is higher to those living in and near cholera cases. Rapid identification of cholera cases and implementation of measures to prevent subsequent transmission around cases may be an efficient strategy to reduce the size and scale of cholera outbreaks.

### Methodology/Principle findings

We investigated implementation of cholera case-area targeted interventions (CATIs) using systematic reviews and case studies. We identified 11 peer-reviewed and eight grey literature articles documenting CATIs and completed 30 key informant interviews in case studies in Democratic Republic of Congo, Haiti, Yemen, and Zimbabwe. We documented 15 outbreaks in 12 countries where CATIs were used. The team composition and the interventions varied, with water, sanitation, and hygiene interventions implemented more commonly than those of health. Alert systems triggering interventions were diverse ranging from suspected cholera cases to culture confirmed cases. Selection of high-risk households around the case household was inconsistent and ranged from only one case to approximately 100 surrounding households with different methods of selecting them. Coordination among actors and integration between sectors were consistently reported as challenging. Delays in sharing case information impeded rapid implementation of this approach, while evaluation of the effectiveness of interventions varied.

### Conclusions/Significance

CATIs appear effective in reducing cholera outbreaks, but there is limited and context specific evidence of their effectiveness in reducing the incidence of cholera cases and lack of guidance for their consistent implementation. We propose to 1) use uniform cholera case

**Data Availability Statement:** This manuscript presents findings from a literature review and qualitative interviews with health professionals. All

sources identified in the literature review are publicly available and are cited in the manuscript, meeting the requirement of data being available in the manuscript itself. Qualitative interview transcripts will be made available upon request to the Johns Hopkins Center for Humanitarian Health (email: humanithealth@jhu.edu) and after a data sharing agreement is signed to ensure appropriate use.

**Funding:** Funds for this research were provided by the Bureau for Humanitarian Assistance, US Agency for Development (https://www.usaid.gov/) Grant number 720FDA19GR00205 (PS, CA, SD, AA). The funders had no role in study design, data collection and analysis, decision to publish, or preparation of the manuscript.

**Competing interests:** The authors have declared that no competing interests exist.

definitions considering a local capacity to trigger alert; 2) evaluate the effectiveness of individual or sets of interventions to interrupt cholera, and establish a set of evidence-based interventions; 3) establish criteria to select high-risk households; and 4) improve coordination and data sharing amongst actors and facilitate integration among sectors to strengthen CATI approaches in cholera outbreaks.

## Author summary

Cholera transmission risk is higher in those living in and near the case household. A set of preventive interventions are implemented in and around case household to reduce cholera transmission. We investigated the implementation of cholera case-area targeted interventions (CATI) using systematic reviews (11 peer-reviewed and eight grey literature) and four case studies in the Democratic Republic of Congo, Haiti, Yemen, and Zimbabwe with 30 key informant interviewees. We found 15 outbreaks in 12 countries where CATI approaches were used. The interventions varied across outbreaks with water, sanitation, and hygiene interventions being more common than those of health. We found different alert systems to trigger interventions, inconsistent criteria to select high-risk households for CATI implementation, and varied team compositions to implement CATI approaches. Coordination and integration among actors and sectors were identified as challenging in many outbreaks, and delays in sharing case information were reported. Evaluation measures varied, few evaluated cholera transmission reduction. We recommend using uniform case definition considering country's capacity to trigger alert, evaluating effectiveness of the various interventions, establishing criteria to select high-risk households, and improving coordination among actors to facilitate integration to aid future cholera-response CATI approaches.

## Introduction

Cholera remains a significant cause of mortality and morbidity, particularly in low-resource, fragile, and humanitarian settings.[1,2] Cholera, an acute bacterial diarrhea, is transmitted by the fecal-oral route and can be prevented with access to safe water, improved sanitation, and protective hygiene practices.[3] Because of both person-to-person and environmental transmission pathways, the risk of cholera infection is higher to those living in or near the case household.[4,5] To interrupt cholera transmission in high-risk environments, preventive water, sanitation, and hygiene (WASH), health, and surveillance interventions are implemented via various delivery models. One approach is termed case-area targeted interventions (CATI), which are spatially and temporally focused and intended to be delivered to the case household and immediate neighbors as soon as possible after case identification. CATIs are often implemented through rapid intervention mechanisms in response to an increase in cholera cases in a given area and are intended to contain cholera outbreaks and are aligned with recommendations in *Ending Cholera–A Global Roadmap to 2030*.[3] While CATIs are used to interrupt cholera outbreaks, differing implementation practices can affect effectiveness including surveillance mechanisms to trigger a CATI, included interventions, methods of determining coverage area, timeliness, and coordination between the implementing actors.

To comprehensively study the CATI implementation process in different settings, we employed a mixed-methods design including reviews of peer-reviewed and grey literature and

retrospective case studies where the CATI approach was utilized to control cholera. As the terms CATI and Rapid Response Teams (RRT) were interchangeably used, and at the time CATI approaches were used, but the term itself was not used, we used a broad definition of CATI to examine the evidence: *cholera case targeted interventions to interrupt transmission of the disease at the household and/or community level*. This flexible definition allowed us to capture evidence of CATIs irrespective of implementation mechanism by the CATI team visiting case households or case and neighboring households) and included interventions (e.g. health, WASH, surveillance). RRT articles that did not employ cholera case targeted response teams or cholera case targeted interventions were not included in this review. In this manuscript, the varied CATI implementation processes were reviewed, and lessons learned presented about context and other implementation factors that affect cholera responses.

## Methods

A mixed-methods approach to study CATI implementation was employed, including: 1) reviews of peer-reviewed journal publications and grey literature published between January 2009 and November 2019; and 2) four retrospective case studies of cholera outbreaks in the Democratic Republic of the Congo (DRC) (2017–2020), Haiti (2010–2019), Yemen (2016–2020), and Zimbabwe(2018–2019).

### Literature review

The Preferred Reporting Items for Systematic Reviews and Meta-Analyses (PRISMA) Guidelines were used to guide the review.[6] A systematic search of grey and peer-reviewed literature was conducted using a combination of controlled vocabulary and keywords for cholera and rapid response interventions (S1 Table).

The search was limited to publications between January 2009 and November 2019; English language publications were included in both searches, in addition to French and Spanish publications in the grey literature search. The grey literature search used key word searches as well as review of various organizational websites including organizations involved in implementing and funding cholera responses and technical and coordinating bodies. In addition, key documents already known to the authors, including those identified by the READY Initiative,[7]. and forward citation tracking were used to ensure the review was as inclusive as possible.

The screening process for peer-reviewed literature was managed using Covidence software. [8] Titles and abstracts were independently screened by two reviewers; in cases of disagreement, a third reviewer was used as a tiebreaker. Eligibility of full texts were assessed by a single reviewer, and if excluded, the reason recorded. For inclusion, articles had to include primary data or be a systematic review and discuss cholera rapid response intervention(s). Grey literature documents were screened by one reviewer and classified as primary (evaluation reports, after action reviews, and case study reports) or secondary (website posts or blogs in which an organization provided a project update). Primary documents were advanced to full data extraction, while secondary documents were only used to map outbreaks.

Both the included peer and grey literature were categorized as reporting on CATI or other rapid response interventions and only CATI interventions were included in this review. Data extraction used a Microsoft Excel template with fields organized under the following categories: outbreak information, rapid response activation, team composition and equipment/supplies, activities implemented, cross-sectoral integration, coordination, data collected, performance/results, and challenges/limitations. Following initial data extraction key articles and documents were re-reviewed and additional data extracted around particular themes during results synthesis.

## Case studies

Retrospective case studies were used to investigate CATI implementation in DRC (2017–2020), Haiti (2010–2019), Yemen (2016–2020), and Zimbabwe (2018–2019) where the approach was implemented to control cholera outbreaks. Locations were selected in consultation with a strategic advisory group that included members from United Nations Children's Fund (UNICEF), World Health Organization (WHO), and United States Agency for International Development. Personal contacts, referrals, and snowball sampling were used to identify potential key informants; the DRC, Haiti, and Yemen case studies had eight to nine key informants, whilst Zimbabwe had four.

A semi-structured interview guide was developed that included outbreak background; decision-making; response actors; flow of information; interventions; coordination and integration of WASH, health, and surveillance activities; change in interventions over time; challenges; and lessons learned. The interviews lasted approximately one hour, and follow-up interviews/correspondence were undertaken as needed; participants were also requested to share relevant documentation (DRC = 8, Haiti = 6, Yemen = 0, and Zimbabwe = 1). After completing interviews, results were compiled by topic and country-level reports prepared; drafts were shared with key informants and feedback incorporated as appropriate in final versions. A non-human subject's research determination was received from the Johns Hopkins Bloomberg School of Public Health Institutional Review Board for the case study component.

## Results

Results from literature review and case studies were synthesized to prepare a comprehensive description of the CATI implementation process and present common implementation challenges. In the systematic review, 1,281 peer-reviewed articles were identified, and 16 peer-reviewed articles were ultimately retained, of which 11 reported on CATIs (10 from direct search result and one from forward citation tracking) and five on other rapid response interventions as outlined in the PRISMA diagram (Fig 1). In the grey literature search, 101 documents were identified and 57 were classified as primary documents. Cholera rapid response interventions were described in 29 grey literatures documents and eight reported on or mentioned CATI implementation. Additionally, one document was included during the peer-review process following one of the reviewer's suggestion. The combined peer-reviewed and grey literature searches yielded 74 documents that were included in the review, of which 20 specifically documented CATIs. A total of 30 key informants from UN agencies (47% of the respondents), non-governmental organizations (NGOs) (23%), government (20%), and academics (10%) involved with the CATI implementation were interviewed between July and October 2020.

## Outbreaks

A total of 15 outbreaks of varying durations in 12 countries were identified, including four from DRC,[9] two from Zimbabwe,[10] and one each from Cameroon,[11] Kenya,[12] Nigeria,[13] Sierra Leone,[14,15] Guinea,[15] South Sudan,[16] Yemen,[10,13,17–21] Haiti, [10,13,22–24] Bangladesh,[25,26] Nepal (Table 1).[27]

Most literature documents focused on a single outbreak with only two addressing multiple outbreaks. Within our literature search period, CATIs were first reported in the 2004 Douala, Cameroon outbreak and use continued through 2020 in the ongoing cholera outbreaks DRC and Yemen case studies.[11,30] The size of outbreaks where CATIs were employed varied widely, ranging from 169 confirmed cases in Kathmandu Valley, Nepal to more than 2.3

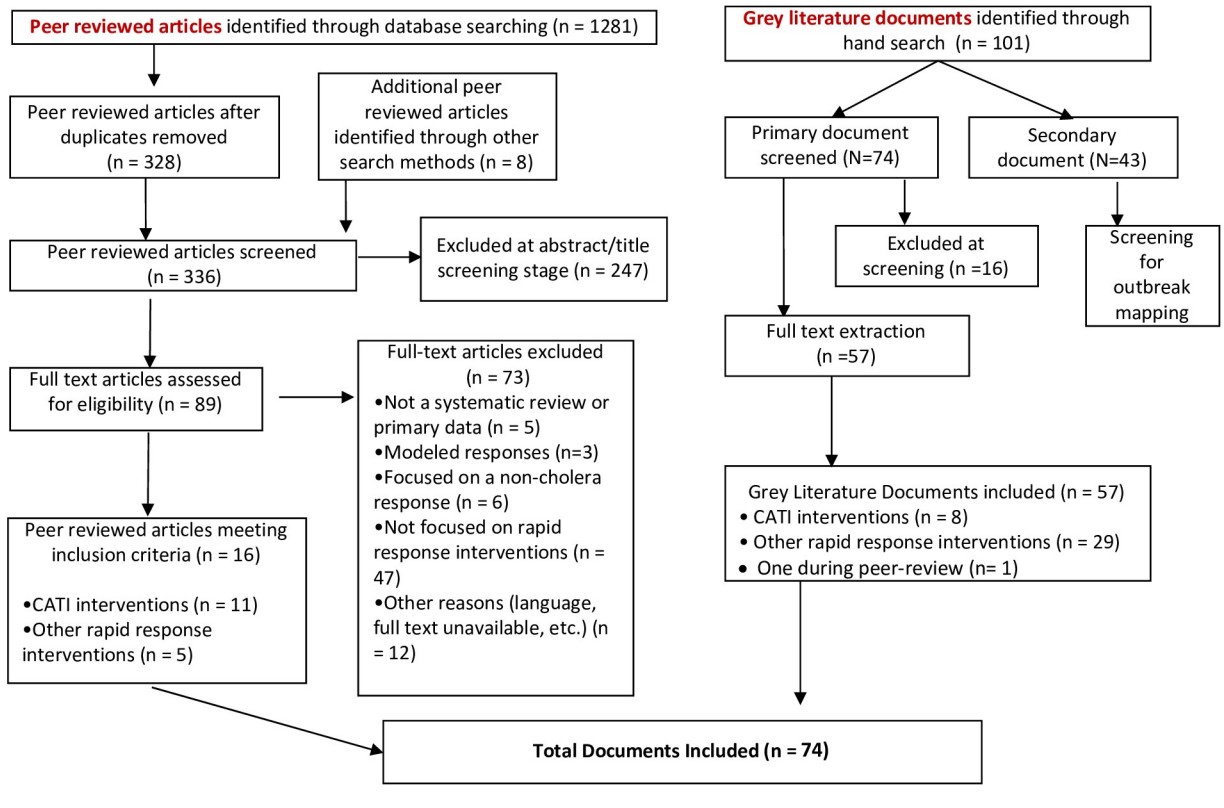

**Fig 1. PRISMA flow diagram.**

million suspected cases in Yemen.[27,31] Additionally, outbreak duration ranged from one month in Juba, South Sudan to over eight years in Haiti;[16] the entire duration and extent of outbreaks was not consistently reported. In Bangladesh, Cameroon, DRC (some of the outbreaks), Kenya, Sierra Leone, and Zimbabwe, cholera occurred in the absence of a humanitarian emergency, whereas in DRC, Nigeria, South Sudan, and Yemen conflict was ongoing and outbreaks in Haiti and Nepal occurred several months after the earthquakes during earthquake recovery. CATIs were implemented in urban settings in Bangladesh, Cameroon, Guinea, and Nepal, and rural settings in Kenya; in the other countries, epidemics occurred in both settings.

## Interventions

CATIs typically included a combination of preventive WASH, health, and surveillance activities to interrupt cholera transmission. We observed that interventions varied across the outbreaks and temporally within the same outbreak, particularly in multi-year outbreaks (Table 2).

From the peer-reviewed literature, CATIs either encompassed WASH only (n = 6) or both health and WASH activities (n = 5); there were no CATI teams delivering health interventions alone. All WASH CATIs included water disinfection at the household or community level and/or hygiene education along with distribution of water treatment supplies. Other interventions provided in only some locations included safe water storage (Haiti and Bangladesh); soap provision (Haiti, Bangladesh, South Sudan, Kenya); and oral rehydration salts (Haiti). Health interventions delivered by CATI teams included antibiotic chemoprophylaxis for case households in Haiti (doxycycline for non-pregnant adults, though delivery was inconsistent)

**Table 1.  Included cholera outbreak locations and reporting dates.**

| | Peer-reviewed* | Grey literature* | Case study* |
|---|---|---|---|
| **African Region** | | | |
| DRC | | Eastern Provinces (2017)[9] | Four outbreaks (2017–2020) |
| Cameroon | Douala (2004)[11] | | |
| Guinea | National (2012)[15] | | |
| Kenya | Nyanza Province (2008)[12] | | |
| Nigeria | | National (2017–2018)[13] | |
| Sierra Leone | | National (2012)[14,15] | |
| South Sudan | Juba (2015)[16] | National (2017–2018)[13] † | |
| Zimbabwe | | National (2008–2009),[28] Harare (2018–2019) [10] | Harare (2018–2019) |
| **Eastern Mediterranean Region** | | | |
| Yemen | National (2016–2018),[17] Hodeidah (2016–2017)[18] | National[10,13,17,19,21,29] | National (2016–2020) |
| **Region of the Americas** | | | |
| Haiti | National (2013–2017),[23] Centre (2015–2017),[22] Port au Prince (2010–2011)[24] | National[10,13] | National (2010–2019) |
| **South-East Asia Region** | | | |
| Bangladesh | Dhaka (2013–2014)[25,26] | | |
| Nepal | Kathmandu Valley (2016)[27] | | |

*Reporting duration is not the same as outbreak duration.

† The report included description of cholera specific RRT, however, they have not responded to a cholera outbreak.

and for both case and adjacent households in Cameroon (doxycycline for the general population and amoxicillin for children and pregnant/lactating women). Oral cholera vaccine (OCV) was delivered as part of CATIs in South Sudan, where there was surplus stock from a previous vaccination campaign. Grey literature contained relatively little information about specific interventions. All documents reported on WASH interventions, including household disinfection, hygiene education, water chlorination, water quality testing, sanitation assessment, and waste management. Health interventions were described in four documents, including antibiotic chemoprophylaxis distribution in Haiti, health education in Sierra Leone and Yemen, health counseling and referrals in Yemen, and case investigation. Variation in CATI implementation was also observed in the four case study countries where WASH interventions were reported in all locations and health and surveillance interventions were reported in DRC, Haiti, and Yemen but not Zimbabwe.

## Team composition

CATI team composition varied across outbreaks and within the same outbreak in Haiti and Yemen. The teams were comprised of staff from NGOs, governments, and community volunteers. Teams were most commonly comprised of four to seven members with a mixed set of skills, including WASH, health/nursing, hygiene promotion, surveillance, and drivers. Teams with only WASH staff were reported in case study interviews in Yemen, Zimbabwe, and during the early response in Haiti;[23] health CATI teams were reported in Yemen.[30] The skill sets included depended upon the type of interventions delivered. For instance, a nurse accompanied CATI teams in Haiti to administer antibiotics and vaccinators were CATI members in

**Table 2. List of WASH, health, and surveillance interventions implemented in CATIs.**

| | DR Congo | Haiti | Yemen | Zimbabwe | Bangladesh | Nepal | Cameroon | Kenya | Sierra Leone | South Sudan | Nigeria | Guinea |
|---|---|---|---|---|---|---|---|---|---|---|---|---|
| **Household Interventions** | | | | | | | | | | | | |
| **WASH** | | | | | | | | | | | | |
| Household disinfection | X | X | X | X | | | X | | | | | X |
| Latrine disinfection | X | X | X | X | | | X | | | | | |
| Hygiene education session | X | X | X | X | X | X | X | X | X | X | X | X |
| Chlorine tablet distribution | X | X | X | X | X | X | | X | | X | | X |
| Water storage container distribution | X | X | X | X | X | X | | | | | | |
| Water collection container/jerrican | X | | X | X | | | | | | | | |
| Water quality testing | X | X | X | X | | X | | | | | | |
| Hygiene promotion flyer | | X | | | X | X | | | | | | |
| Soap distribution | X | X | X | X | | | X | | X | | X | |
| **Health** | | | | | | | | | | | | |
| Antibiotic chemoprophylaxis | X | X | | | | | X | | | | | |
| Oral rehydration salt distribution | X | X | X | | | | | | | | | X |
| Oral cholera vaccine | | | | | | | | | | X | | |
| **Surveillance** | | | | | | | | | | | | |
| Case identification at treatment center | X | X | X | X | | | | | | | | X |
| Active case finding | | X | X | | | X | | | | | | |
| Referrals to CTC | | X | X | X | | | | | | | | X |
| **Community and health facility-based interventions** | | | | | | | | | | | | |
| **WASH** | | | | | | | | | | | | |
| Health promotion | X | X | X | X | | | X | | X | | | X |
| Point-of-use disinfection product distribution | | | | X | | | | | | | | |
| Bucket chlorination | X | X | | | | | | | | | | |
| Chlorination at water point | X | X | | | | | | | X | | | X |
| Latrine construction in public areas | X | | | | | | | | | | X | |
| Water point rehabilitation | X | X | | | | | | | | | X | X |
| Safe burials | | X | | | | | | | | | | X |
| Community volunteer training | | X | | | | | | | | | | |
| **Health** | | | | | | | | | | | | |
| Antibiotic chemoprophylaxis | | X | X | X | | | | | | | | |
| ORT* through mobile clinics | | X | X | X | | | | | | X | | |

Note: Interventions are reported as implemented in a country if conducted at least in one outbreak, or for a period of time in the case of long outbreak). Therefore, variability within country over space and time is not captured in this table.

* ORT = oral rehydration therapy; CTC = cholera treatment center

South Sudan to provide OCV. Additionally, the Haiti case study showed that before establishing the mixed teams, comprised of health and WASH staff, the NGO WASH teams would often include a nurse to accompany them.

## Implementation

CATI implementation was a multi-faceted process involving numerous steps and multiple organizations typically beginning with an alert of a suspected/confirmed cholera case(s) (Fig 2). In Nepal,[27] Bangladesh,[26] and South Sudan,[16] CATIs were implemented in response

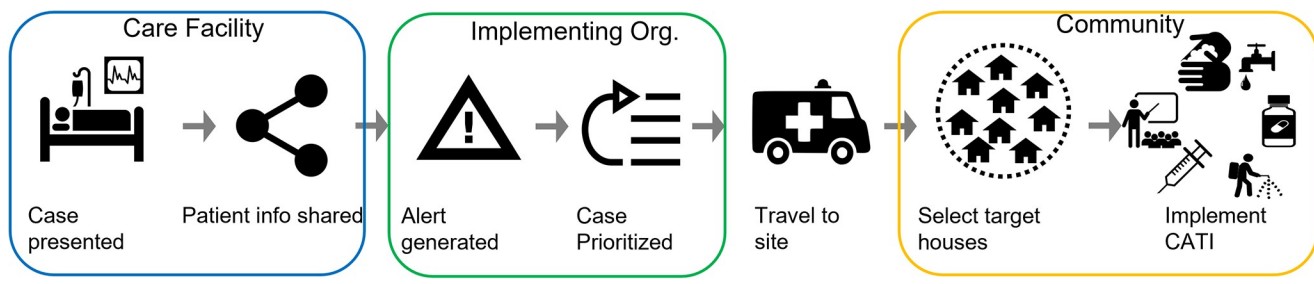

**Fig 2. Common steps in the CATI implementation process.**

to rapid diagnostic test (RDT) or culture confirmed positive cases; both were used at different times in Yemen. [17]However, in Cameroon,[11] DRC,[10] and Zimbabwe,[32] CATIs were implemented for suspected cases that presented for care at a cholera treatment center. RDT was used in Haiti during 2013 and discontinued afterward because of delays to obtain results and reliability concerns. Specific CATI activation alert systems were not reported in Nigeria, Kenya, and Sierra Leone. The common source of information to generate an alert was the shared line list from the health treatment facility.[13,20–23] Additionally, suspected cases reported by community health workers, media reports, and other informal surveillance mechanisms also triggered alerts in DRC,[9] Haiti, and Yemen.[13] The time between the identification of a suspected case and the trigger of an alert was critical for rapid implementation of CATIs and community level alert systems were used to prioritize response.[23] Delays in sharing case information with CATI teams were frequently reported.[13,20–23] For instance, in Yemen, cholera case line lists had to be approved by central authorities before being shared [13,17] and in Haiti, case information was shared only on a weekly basis during the initial response which delayed the CATI activation. In Nepal, delays occurred as CATIs were implemented only after receiving a positive stool culture, which took an average of 3.9 days following hospital admission.[27] In Bangladesh, DRC, and Haiti, after establishing daily line lists access, and in South Sudan and Zimbabwe, the CATI teams were able to respond more quickly than in the aforementioned locations[16,22,25,26] as they had rapid access to the case information. In instances when the number of suspect/confirmed cases exceeded the capacity of the CATI teams, a set of prioritization criteria were used, including number of cases from one location (Haiti, Zimbabwe), areas without recent cases (Haiti, Zimbabwe), if a case was RDT/culture positive (Yemen), and case severity status/death (Haiti). [13,22,23]

When CATI teams had difficultly locating case households, assistance was often sought from community members. The selection process and the number of surrounding households to be included in the CATI varied greatly across outbreaks, and depended on the density and arrangement of houses, implementation strategy, capacity, and resource availability. For instance, in Bangladesh and South Sudan, only household contacts received the interventions whereas the early WASH CATIs in Yemen included about 100 households and in the later Yemen response and Haiti a 50–100 meter radius for household selection was reported;[13] in the Zimbabwe case study, the entire apartment level was included when responding in urban settings.[30] As mentioned earlier, a combination of preventive WASH, health, and surveillance activities are used to interrupt cholera transmission, with the same or different interventions provided to case and neighboring households. For example, in Haiti antibiotic chemoprophylaxis was only administered to household contacts;[22] and latrine disinfection was conducted only at case households in Haiti, Zimbabwe, and in Cameroon.[11] In addition to case- and neighbor-targeted household-specific interventions, CATIs in DRC,[9] Sierra

Leone and Guinea,[15] Yemen,[18] and Zimbabwe (case study) included community-level activities such as water point repair, WASH risk factor assessment, testing of communal water sources, construction of communal sanitation facilities, and awareness raising.

## Implementation challenges

Several common difficulties were identified including: 1) implementation in logistically challenging contexts[13,17,23,27] and poor access to targeted populations;[9,12,16] 2) cholera cases exceeding the limited capacity to implement all CATIs;[10,13,22,27] 3) coordination among actors and integration of WASH, health, and surveillance activities;[10,13,15,19] 4) timely access to case information to rapidly implement CATI;[10,23] and 5) limited funding to adequately implement the interventions.[9] The grey literature review and all four case studies reported coordination between WASH and health actors as hard to achieve, at times resulting in duplication of activities by the WASH and health CATI teams. Aspects such as access to case data as well as coordination among partners gradually improved over time, which enhanced response timeliness in Haiti[23] as well as in Yemen.[30]

## Evaluation measures

Multiple measures were used to evaluate different aspects of CATI performance including: 1) timeliness (e.g. time between case presentation/confirmation and CATI start or time between outbreak declaration/index case confirmation and start of the first CATI); 2) coverage; 3) cost of the intervention; 4) household's knowledge and practice of measures aiming to interrupt cholera transmission; 5) outputs (items distributed during intervention); and 6) impact (incidence of cases). While the Bangladesh and Haiti studies reported about the complete CATI approach, none of the documents reviewed evaluated the effect of individual interventions (e.g. only chlorine tablet distribution or only awareness session) in reducing the number of new cases or interrupting the cholera transmission pathways.

Timeliness of intervention delivery was reported in four peer-reviewed[16,22,23,27] and two grey literature documents.[10,13] The elapsed time between case admission and the start of CATIs varied across outbreaks, and usually CATI teams based at cholera treatment facilities were able to initiate interventions faster than separate organizations not based in the facilities. In Cameroon,[11] DRC,[33] Haiti,[24] and Bangladesh[26] the interventions started at case admission or discharge where antibiotic prophylaxis and hygiene kits were distributed to the case's household. In South Sudan, Nepal, and Zimbabwe, interventions at the case household began between one and four days after case identification. In Nepal, delivery started on average 1.7 days after culture result and 3.9 days after admission,[27] and in Zimbabwe, the intervention started within 48 hours of the presentation of a suspected case at the treatment center,[9] and in South Sudan, CATIs occurred 1–6 days after the suspected case was presented at the health facility with an average delay of 3.4 days.[16]

Intervention coverage was reported in 10 peer-reviewed articles and four grey literature reports, though different indicators were used. For example, 51% cases in South Sudan[16] and 92% in DRC received interventions;[9] whereas 17% and 30% of households and communities, respectively, received interventions in Nepal.[27] In Haiti, coverage was reported in terms of responded cholera alerts (49%),[23] identified outbreaks responded (53% in Centre department)[22] and cases admitted to cholera treatment centers in Carrefour Haiti (65%). [24] The two Bangladesh research studies reported 100% coverage of the admitted cases, which is not typical in most outbreak responses. One peer-reviewed article and two grey literature documents reported CATI cost information using different metrics. In Sierra Leone the per capita cost was US$2.32 (and was compared with a previous outbreak response in

Zimbabwe where the cost was US$2.85)[14] and in Bangladesh the cost was US$45.5 per household.[26] UNICEF's global review of cholera rapid response teams reported monthly per team costs of $10,234 in Haiti and $2,400 (urban) to $3,000 (rural) in Yemen.[13]

A range of different effectiveness measures were reported in peer-reviewed articles. Behavior change measures, such as knowledge of cholera prevention, treatment, and hygiene practices;[12,25] and use of appropriate hygiene practices and/or items distributed in hygiene kits[26,27] were commonly reported. Only the Bangladesh study and the Haiti Central department study reported evaluated differences in cholera incidence between the households that received a timely CATI and those that did not.[22,26] For instance, in Kenya, a significant difference in respondent-reported water treatment as compared to controls (56% vs 37%, p<0.001) was reported, however, cholera knowledge (7/7 indicators) and behaviors (5/6 indicators) were similar between the two groups.[12] In Bangladesh, as compared to a control group, the intervention group had significantly higher odds of handwashing (odds ratio = 14.7, CI: 8.3–25.9); was more likely to have soap present in cooking areas (98% vs. 15%, p<0.001) and latrine areas (98% vs 13%, p< 0.0001); stored drinking water was more likely to have adequate concentrations of free chlorine (94% vs 1%, p<0.0001) and less likely to be contaminated with *V. Cholerae* (0% vs 6%, p = 0.06); and there was a 47% lower incidence of *V. cholerae* infection (symptomatic and asymptomatic) in intervention group case contacts during the intervention period.[25,26] In Nepal, 30.2% of surveyed households reported hearing awareness messaging via community campaigns and 16.5% reported home visit with various types of health education messages and supplies. Individuals that received home visits were more likely to have heard of cholera (adjusted odds ratio = 2.38, p<0.05).[27] In Haiti, estimates of the effectiveness of national CATIs indicated that timely response (≤1 day after outbreak identification compared to a delayed response of >7 days) reduced new cases by 76% (CI: 59–86%) and outbreak duration by 61% (CI: 41–75%) whereas an intense response (≥1 complete CATI per week compared to a weaker response of ≤0.25 complete CATIs weekly) reduced new cases by 59% (CI: 11–81%) and outbreak duration by 73% (CI: 49–86%).[22]

## Discussion

Our mixed-methods research synthesized the CATI implementation approach from 15 different outbreaks in 12 countries. We noted many ambiguities and inconsistencies regarding RRTs and CATIs in terms of definitions, interventions combinations and implementation strategies, integration, coordination, data sharing, and reporting across outbreaks. CATI approaches were used to reduce cholera transmission in different settings (e.g., urban, rural) and outbreaks of differing scale and duration. The CATI approach was used as a nationwide strategy in Haiti and Yemen, and was incorporated into the national cholera control strategy in DRC,[34] South Sudan,[35] and Nigeria.[13]

The cholera case definition (i.e. suspected, RDT positive, and culture positive) to initiate a CATI implementation varied across outbreaks based on contextual factors such as laboratory and procurement capacity, and size of the outbreak. The use of an alert system with specific mechanisms to trigger a CATI response was described in eight countries. While the specific mechanisms varied across outbreaks, the alert systems facilitated a rapid response. The use of a national case definition that considers existing surveillance and laboratory testing capacities, and links those with an alert system to trigger CATI response can strengthen early detection and quick response to contain outbreaks. The Global Task Force on Cholera Control (GTFCC) has proposed suspected and confirmed case definitions in outbreak and non-

outbreak settings that can be used in the absence of national case definitions to trigger a CATI response.[3]

CATIs typically included a varied set of WASH, health, and surveillance interventions that were implemented at the case and/or neighboring households, and in affected communities (Table 2); however, the interventions were not consistently implemented across outbreaks. WASH activities such as the distribution of water disinfection supplies, water storage containers, and hygiene education predominated. While all CATIs included surveillance activities, in Bangladesh, Nepal, Kenya, Sierra Leone, and Zimbabwe health interventions at the household level were not provided. However, with the exception of the Bangladesh study, which reported three months of piloting and previous formative research to design the intentions, selection of interventions was not reported. [26] Establishing a set of specific and standardized WASH, health, and surveillance interventions according to context and that are considered to be effective to interrupt cholera transmission would support implementers in selecting activities that are relevant to the outbreak setting.

The scientific evidence of the effectiveness of CATIs in reducing cholera transmission is limited and context specific. A recent scoping review reported evidence of some interventions CATI activities such as antibiotic chemoprophylaxis, single-dose OCV, intensive hygiene promotion, and point-of-use water treatment to rapidly limit cholera transmission in the household and its high-risk radius.[36] Several measures of CATI effectiveness were identified in our study, including timeliness, completeness, coverage, and implementation cost. While these criteria provided information on strength of CATIs implementation, they did not measure impact in terms of reduced incidence of cholera. The exception were studies in Bangladesh, which reported cholera infections in control and intervention groups, and Haiti which reported the relationship between response speed and intensity and outcomes of case reduction and outbreak duration.[22,26] Additional research on the potential of CATIs to reduce cholera case incidence in different transmission settings and on the contribution of the individual interventions to case reduction would be valuable. Yet, the complexity of measuring and attributing effectiveness to one approach or one intervention during an outbreak in general and even more so in humanitarian settings should be recognized. Challenges include but are not limited to the identification of a control group; the need to adapt the response to the context, which makes comparisons difficult; delays in how rapidly research can start; and concerns around sample sizes and power of the studies. One potential alternative which would expand the breadth of evidence is strengthen monitoring systems of responding organizations to be able to better document the response, and to work towards the adoption of standardized indicators to enable comparison of evidence across outbreaks. Additionally, partnership between research institutions and emergency responders to systematically study the different components of the CATI implementation can also improve the reliability and generalizability of the evidence.

The selection criteria for households included in CATIs varied between including only case households to approximately 100 surrounding households. A recent review reported a 100m radius around the case as appropriate but proposed further study of implementation feasibility in urban settings.[36] Additionally, a study in Kinshasa, DRC reported the use of a targeted grid approach delivering WASH interventions to contain the outbreak in urban settings.[37] The feasibility of using a certain number of households as compared to a ring may depend on settings. For instance, in densely populated areas and multistoried buildings, using number of households to determine CATI coverage is easier to implement than a distance radius-based approach. While establishing fixed numbers of households or a radius is difficult considering the heterogeneity in dwelling arrangements in urban and rural settings, evidence-based

guidance for neighbor household selection considering different settings could facilitate CATI delivery and potentially increase CATI effectiveness.

This review aimed to investigate the integration of WASH, health, and surveillance activities in CATI, both in terms of level of integration as well as the possible positive effects on the overall performance of CATI. Integration was defined in terms of conducting activities from the three sectors and including different technical profiles in the same team. While in few instances only WASH interventions were delivered, we found that in most of the cases, CATI were integrated, and included WASH, health, and surveillance activities. Considering the time and resources needed to implement CATIs, it may seem more efficient in outbreak settings to integrate the interventions so that they can be implemented by one CATI team. However, the evidence on the effectiveness of integrated versus individual WASH and health CATIs is mixed. Further research on the integration of health interventions in CATIs is of strategic importance and should be further explored due to their potential to limit transmission (as demonstrated in non-CATI contexts).[36]

Operational challenges to coordinate between the actors were frequently reported, which can reduce the effectiveness of a joint intervention. CATI implementation often involved multiple actors (e.g. government, UN, NGOs) and both WASH and health sectors. Functioning coordination between the actors and sectors can improve implementation. For instance, direct access to cholera patient line lists from treatment facilities was limited and delayed in multiple responses. The usefulness of CATIs heavily relies on early detection of an outbreak and rapid response to interrupt person-to-person and environmental transmission.[4] Therefore, it is imperative for all involved actors (authorities, health facilities, and CATI implementers) to establish an effective coordination mechanism that facilitates rapid access to cholera case information for CATI teams.

Our peer-reviewed and grey literature search was limited to the past decade and peer-reviewed articles were limited to the English language. This may have reduced our capacity to identify relevant CATI literature that was published earlier; however, inclusions of documents identified via forward citation tracking, irrespective of publication date, mitigates this concern. While extensive, the grey literature did not provide the level of detail reported in the peer reviewed literature. Therefore, certain elements related to CATI implementation may not have been fully reported, especially as none of the documents focused solely on CATI. Our case studies included CATI experiences from four countries; inclusion of additional key informants, in particular government representatives, for case study countries and expanding the number of countries and contexts could have strengthened our findings and provided additional perspectives about the implementation process. Majority of the countries in the study were deemed to be fragile by the World Bank. [38] While studying the implications of countries fragility on the effectiveness of CATIs were beyond the scope of the manuscript, readers should consider this caveat when interpreting the results.

To better characterize the CATI approach and differentiate it from other response modalities, we suggest defining CATI as cholera case targeted interventions to interrupt transmission of the disease at the household and/or community level. Future studies and operational reports should provide precise intervention descriptions to differentiate CATI more easily from other rapid response mechanisms that are not focused on the individual case.

## Supporting information

**S1 Table. Literature search strategy summary.**
(DOCX)

## Author Contributions

**Conceptualization:** Mustafa Sikder, Chiara Altare, Daniele Lantagne, Andrew S. Azman, Paul Spiegel.

**Data curation:** Mustafa Sikder, Chiara Altare, Shannon Doocy, Daniella Trowbridge, Paul Spiegel.

**Formal analysis:** Chiara Altare, Paul Spiegel.

**Funding acquisition:** Paul Spiegel.

**Investigation:** Mustafa Sikder, Shannon Doocy, Daniella Trowbridge, Gurpreet Kaur, Natasha Kaushal, Emily Lyles, Daniele Lantagne, Paul Spiegel.

**Methodology:** Mustafa Sikder, Chiara Altare, Shannon Doocy, Gurpreet Kaur, Emily Lyles, Andrew S. Azman, Paul Spiegel.

**Project administration:** Chiara Altare, Paul Spiegel.

**Supervision:** Daniele Lantagne, Paul Spiegel.

**Validation:** Mustafa Sikder, Chiara Altare, Paul Spiegel.

**Visualization:** Mustafa Sikder.

**Writing – original draft:** Mustafa Sikder, Paul Spiegel.

**Writing – review & editing:** Mustafa Sikder, Chiara Altare, Shannon Doocy, Daniella Trowbridge, Gurpreet Kaur, Natasha Kaushal, Emily Lyles, Daniele Lantagne, Andrew S. Azman, Paul Spiegel.

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
