## [Decision Letter · Decision Letter 0]

3 Sep 2021

Dear Dr. Spiegel,

Thank you very much for submitting your manuscript "Case-area targeted preventive interventions to interrupt cholera transmission: current implementation practices and lessons learned" for consideration at PLOS Neglected Tropical Diseases. As with all papers reviewed by the journal, your manuscript was reviewed by members of the editorial board and by several independent reviewers. In light of the reviews (below this email), we would like to invite the resubmission of a significantly-revised version that takes into account the reviewers' comments. 

We cannot make any decision about publication until we have seen the revised manuscript and your response to the reviewers' comments. Your revised manuscript is also likely to be sent to reviewers for further evaluation.

Sincerely,

Jade Benjamin-Chung

Guest Editor

Emily Gurley

Deputy Editor

Reviewer's Responses to Questions

**Key Review Criteria Required for Acceptance?**

**Methods**

-Are the objectives of the study clearly articulated with a clear testable hypothesis stated?

-Is the study design appropriate to address the stated objectives?

-Is the population clearly described and appropriate for the hypothesis being tested?

-Is the sample size sufficient to ensure adequate power to address the hypothesis being tested?

-Were correct statistical analysis used to support conclusions?

-Are there concerns about ethical or regulatory requirements being met?

Reviewer #1: Are the objectives of the study clearly articulated with a clear testable hypothesis stated? YES

-Is the study design appropriate to address the stated objectives? YES

-Is the population clearly described and appropriate for the hypothesis being tested? NO. Inclusion of studies on Rapid Response Team as equivalent to CATI was not appropriate. YES

-Is the sample size sufficient to ensure adequate power to address the hypothesis being tested? YES

-Were correct statistical analysis used to support conclusions? YES

-Are there concerns about ethical or regulatory requirements being met? YES

Reviewer #2: - Yes, the objectives of the study are clearly articulated

- Yes, the study design is appropriate 

- The analyzed literature used for this study should better be described, and choice of analyzed papers and grey literature better explained (see comments bellow)

- Some key articles and reports are missing in the review (see comments bellow)

This study includes no statistics and present no experiments

**Results**

-Does the analysis presented match the analysis plan?

-Are the results clearly and completely presented?

-Are the figures (Tables, Images) of sufficient quality for clarity?

Reviewer #1: Does the analysis presented match the analysis plan? YES

-Are the results clearly and completely presented? YES

-Are the figures (Tables, Images) of sufficient quality for clarity? NO, Table need revision

Reviewer #2: - Characteristics of selected articles and reports should better be presented (see comments bellow)

- Part of the results should better be presented (see comments bellow)

- yes, figures are of sufficient quality

**Conclusions**

-Are the conclusions supported by the data presented?

-Are the limitations of analysis clearly described?

-Do the authors discuss how these data can be helpful to advance our understanding of the topic under study?

-Is public health relevance addressed?

Reviewer #1: Are the conclusions supported by the data presented? YES

-Are the limitations of analysis clearly described? YES

-Do the authors discuss how these data can be helpful to advance our understanding of the topic under study? YES

-Is public health relevance addressed? YES

Reviewer #2: - Part of the conclusions is not supported by the data presented (see general comments bellow)

- Limitations of the study are not addressed in the Discussion

- Yes, authors discuss how these data can be helpful, and public relevance is addressed

**Editorial and Data Presentation Modifications?**

Reviewer #1: For the purpose of reviewing the contents of the manuscript, it will be more easier for the reviewers if the authors included line numbering on each of the sentences.

Reviewer #2: Abstract

- I recommend against citing the first reference from Ali et al. even if this paper has been widely cited. Indeed, it provides doubtful estimates of the global burden of cholera based on unreliable extrapolations from only 3 limited cross-sectional cholera prevalence surveys ! I suggest authors cite yearly WHO records even though they are likely underestimates. 

- In their conclusion, authors write that CATI « are believed to be effective in reducing cholera outbreaks, but there is

insufficient evidence in their effectiveness ». I do not agree with this statement as 

Authors Summary:

- in « Evaluation measures varied, however, few evaluated cholera transmission reduction » : remove « however » ?

Introduction:

- Authors should rather cite the number of cholera suspected cases recorded by WHO rather than this 

Methods

- Rather than citing the Cochrane reference [5], authors could directly cite the PRISMA 2020 statement of the EQUATOR Network (https://www.equator-network.org/reporting-guidelines/prisma/)

- The PRISMA checklist is missing : https://www.equator-network.org/reporting-guidelines/prisma/

Results

- The process of outbreak selection is not clear to me. Did authors only select outbreaks they found CATI reports on ? Please clarify

- Similarly, I do not understand whether the references cited in Table 1 only intend to describe the listed epidemics, or the CATI implemented strategies. In the latter case, and for example for Haiti, Table 1 could directly include ref [21] and [27] 

- For RDC, authors should include the report from Bompangue et al 2020 (https://doi.org/10.1186/s12879-020-4916-0). 

- For Sierra Leone and Guinea in 2012, authors should include this report: https://plateformecholera.info/attachments/article/394/5-GCSL_2012_SLL_ACF_001.pdf

- A first description of cholera rapid response teams was published in 1971 by Voelkel (http://www.ncbi.nlm.nih.gov/pubmed/5576843)

- A CATI strategy was also implemented in 1998-1999 in Comoros with support of the Medecins du Monde NGO (https://devsante.org/articles/l-epidemie-de-cholera-aux-comores) 

- Page 9, ref [14] was published in 2017 and does not describe 2013-2019 CATI strategy in Haiti. Authors could rather cite ref [21] or [27] , or a more recent letter (https://doi.org/10.1016/s2214-109x(20)30430-7). 

- Please note that the study [20] describes a limited strategy implemented in Port-au-Prince in 2010-2011, which was not part of the nationwide CATI strategy launched in 2013. Further citations of this ref [20] are thus often ambiguous.

- page 10: please note that although the Haiti epidemic occurred 10 months after the earthquake, the disease was imported by a UN peace keeping battalion in a rural area (Mirebalais commune) and exploded 80km downstream of a highly likely contaminated river in a rice field plain, both of which had not been affected by the earthquake at all. Please modify the sentence as it wrongly suggests that cholera may have been triggered by the earthquake. 

- Similarly, are authors sure that the 2016 cholera epidemic in Nepal was influenced by the April 2015 earthquake ?

- page 13: there was no vaccinator in the CATI teams in Haiti as no OCV was administered (Table 2). Please add « respectively »

- page 13: please explain what were the mixed teams in Haiti

- page 14 : as described in ref [27], a specific weekly alert system at the communal level was initiated in Haiti in 2013 in order to help prioritize and monitor CATIs. This system preceded the generalization of case line list sharing. Authors should add this information. 

- page 14: in Haiti, a nation-wide use of RDT was implemented from 2013 in order to help prioritizing CATIs. This strategy was later abandoned as RDT results were not readily available for RRTs and their results proved unreliable. 

- page 14: results from stool culture were also used for CATI implementation in Haiti during the last years of the strategy: as part of the elimination process, RRTs were asked to implement a second CATI in case of positive culture

- page 15: please add references to « 4)timely access to case information to rapidly implement CATI ». For instance ref [27] was cited for the same topic on page 14.

- page 17: in Haiti, coverage was also reported as the proportion of responded cholera alerts (see ref [27] : 49% between July 2013 and June 2017). 

- page 17: the UNICEF’s global review of cholera rapid response teams reports a monthly cost of $10,234 per team in Haiti (not $180!), and $2,400-3,000 in Yemen (not $1,776). Please check and correct.

- in « differences in cholera incidence between those that received a timely CATI », the term « those » is imprecise.

- page 16-18: I recommend to present behavior change results (that can be considered as CATI outcomes), before effectiveness and impact results. The global « theory of change » scheme could be used to better arrange this part of the review (see for instance http://fic.tufts.edu/assets/WASH-Systematic-Review.pdf)

Discussion

- page 19 : I do not understand the meaning of « which reported formative research to design the intentions ». I do not get the specificities of this very good study [23] on that aspect.

- page 22: considering the proposed definition of CATI, do authors consider that interventions targeting a cholera cluster (e.g. a neighborhood, a locality or even a commune) could be considered as a CATI ?

- no paragraph on the limitations of the study

References :

- ref [8]: URL does not seem to work

- ref [18]: URL does not work

- ref [19]: URL does not work

- ref [20]: URL does not work

- ref [25]: URL does not work

- ref [28]: URL does not work

- ref [31]: URL does not work

**Summary and General Comments**

Reviewer #1: Reviewers’ report for the manuscript PNTD-D-21-00779, Case-area targeted preventive interventions to interrupt cholera transmission: current implementation practices and lessons learned

General comment

This is well written manuscript on a valuable study that synthesizes available literature on CATI as a strategy for prevention and control of cholera outbreaks. Most importantly, the authors investigated and described weaknesses with previous implementation of CATI and give evidence based recommendation to strengthen CATI intervention implementation. The study has the potential to increase scientific knowledge on CATI. However, there are still few important weaknesses in the content that need to addressed and are essential to improve clarity of the message, remove ambiguity in the understanding by the readers and enhance the quality of the contents therein.

The major flaws with this manuscript are in the categorization and inclusion of the studies using the term RRT as synonymous to CATI interventions. Yet CATI and RRT are two distinct terms that are used differently by the cholera outbreak responders/ stakeholders. Also, given that seven out of eleven countries in this paper (table 2.) were from sub-Saharan Africa where cholera prevention is guided by the implementation of Integrated Disease Surveillance and Response (World Health Organization. Technical Guidelines for Integrated Disease Surveillance and Response in the WHO African Region THIRD EDITION. 2019. Available: https://apps.who.int/iris/bitstream/handle/10665/325015/WHO-AF-WHE-CPI-05.2019-eng.pdf) use of RRT as synonymous to CATI has to be very cautious. According to IDSR/WHO, A Rapid Response Team (RRT) is composed of experts who take the lead in conducting the initial investigation of reported and suspected cases or outbreaks so as to confirm the nature of the event under investigation. It is also the responsibility of the RRT to initiate the preliminary control/containment measures needed to prevent further spread of the disease. The same information is also articulated in the cholera prevention WHO RRT guide (World Health Organization. Guide for Rapid Response Teams (RRTs) for Cholera Outbreak investigation & initial response. 2010. Available: https://plateformecholera.info/attachments/article/672/RRT%20%20cholera%20outbreak%20investigation%20guide.pdf). In summary RRT is constituted by experts and there role is to do the initial investigation and response and can be in place before, during and after the cholera outbreak. While CATI is one of the cholera control strategy that can be used by the RRT to contain/interrupt the outbreak. Therefore, Mustafa et al, should revise manuscript and exclude studies on RRT. 

Specific comments

The following comments are related to above concerns. 

1. Introduction. Page 4, second paragraph, “As the terms CATI and Rapid Response Teams (RRT) were interchangeably used, and at the time CATI approaches were used, but the term itself was not used, we used a broad definition of CATI to examine the evidence: cholera case targeted interventions to interrupt transmission of the disease at the household and/or community level” according to IDSR/WHO the term CATI and RRT mean different things. This has the potential to confuse the reader who have accessed IDSR/WHO guidelines. The authors will need to revise this section and possibly exclude studies with RRT from the manuscript.

2. Method section. Page 6, second paragraph, “ Both the included peer and grey literature were categorized as reporting on CATI or other rapid response interventions). This statement is related to above on RRT. Furthermore, other rapid response interventions is open to speculation and misinterpretation. Hence, Mustafa et al, will need to revise this section to remove possible confusion of the readers.

3. Result section. Page10, Table 2. The authors use the terms health and surveillance to categorized activities of CATI. However, the activities listed in surveillance can still fit in health. The authors could therefore merge the two under one sub-heading. 

4. Discussion. Page 18, first paragraph, “ We noted many ambiguities and inconsistencies regarding RRTs and CATIs in terms of definitions, interventions combinations and implementation strategies, integration, coordination, data sharing, and reporting across outbreaks”. The author appreciates that there were many ambiguities for RRTs and CATIs. However, in the subsequent discussion the authors only focus on CATIs and nothing is given on RRTs. This is yet another important reason for Mustafa et al, to revise the contents of the paper in the introduction and methodology section and focus on CATIs as reflected in the title, abstract, discussion and conclusion of this manuscript.

5. Discussion. Page 20, first paragraph, “The scientific evidence of the effectiveness of CATIs in reducing cholera transmission is limited. A recent scoping review found some evidence that antibiotic chemoprophylaxis, single-dose OCV, intensive hygiene promotion, and point-of-use water treatment to rapidly limit cholera transmission in the household and its high-risk radius”. It is clear what message the authors wanted to share. The first sentence points to lack or limited evidence while the second sentence show that there is evidence for effectiveness. Hence, the authors will need to revise this paragraph. 

Other comments

Page 15, paragraph 2, “ Haiti,[20] and Bangladesh[23] the interventions started at case admission or discharge where, for example, antibody prophylaxis and hygiene..” The authors should review the highlighted word and if necessary replace it. 

Page 16, some abbreviations such as CTC are not explained. The authors should review the manuscript and ensure that all abbreviations are explained to avoid misinterpretation by the readers. 

End

Thank you.

Reviewer #2: In this very interesting mixed-methods study using a review of published peer-reviewed and grey literature, authors address the implementation of case-area targeted interventions against cholera.

This piece of work is important as such cholera control strategy has been implemented in several major cholera epidemic over the past decade.

The manuscript is well written, the amount of aggregated information is important, authors propose highly relevant comments and several valuable recommendations, and they should be acknowledged for this piece of work.

Nevertheless, the manuscript suffers from the absence of several important publications and reports, and from several imprecisions and discrepancies which are listed in my specific comments and should be corrected.

Besides, I do not support one aspect of the overall conclusion of this review, which on several occasions implies that CATI effectiveness has not yet been established and that promotion of CATI would so far only be a matter of expert opinion. 

Authors do cite a randomized control trial of hospital-based hygiene and water treatment intervention in Bangladesh [23] which showed that distributing cholera kits in hospital to the family of cholera cases reduced secondary cases. They also cite a quasi-experimental study conducted in Haiti [21], which found that prompt and repeated CATIs were associated with a significant reduction of cholera outbreak duration and cumulative case incidence. Besides, as authors wrote in a recent report of the John Hopkins University about the same subject as the present article (http://hopkinshumanitarianhealth.org/assets/documents/RRT_CaseStudy_Report_2021.pdf), no cholera case has been confirmed in the Haiti since February 2019, which could represent a relevant proof of concept of the CATI approach, as this was implemented as a massive national strategy between 2013 and 2019, with no alternative explanation of cholera disappearance so far. Therefore, they should explain why they do not seem to consider this as as evidence of CATI effectiveness.

Following a modern science-based approach (https://sciencebasedmedicine.org/about-science-based-medicine/), I thus invite authors to consider a less stringent definition of evidence production in such epidemic contexts than the sole randomized control trials. Water disinfection, hand washing, sanitation and so on have for a long time now been shown to prevent fecal-oral diseases such as cholera. 

I thus strongly believe that the state of « clinical equipoise » is not present regarding the question of conducting a CATI or not, and all cases should now be targeted. As a matter of fact, many experts, field public health actors as well as political or administrative leaders do consider that randomizing which cholera case family should or should not receive a CATI during cholera epidemics raises serious ethical issues. 

Nevertheless, I strongly agree with authors that « The scientific evidence of the effectiveness of CATIs in reducing cholera transmission is limited » (page 20). I can only join them on the fact that « measuring and attributing effectiveness to one approach or one intervention during an outbreak » is highly complex. And I totally agree that « monitoring systems of responding organizations » should be strengthened: this is precisely what was performed in Haiti. Some additional evidence to optimize the composition of the CATI package, or the selection of neighboring households to be targeted is urgently needed, and I imagine that valuable randomized studies could be conducted to answer these questions. 

I thus think that authors should modify several sentences using a less ambiguous wording such as :

- conclusion of the abstract: « CATIs are believed to be effective in reducing cholera outbreaks, but there is insufficient evidence in their effectiveness ».

- introduction: « CATI is conceptualized as an efficient way to interrupt cholera outbreaks… »

- Evaluation measures in results: « None of the documents reviewed evaluated the effect of individual interventions in reducing the number of new cases or interrupting the cholera transmission pathways ». This is wrong, as ref [23] demonstrated that distributing cholera kits in hospital to the family of cholera cases reduced secondary cases in Bangladesh, and ref [21] demonstrated that prompt and repeated CATIs were associated with a significant reduction of cholera outbreak duration and cumulative case incidence. 

- Similarly within the same sentence « there were no prospective evaluations of CATI’s effectiveness » : ref [23] was a prospective randomized control trial. And ref [21] was a quasi-experimental evaluation of CATI’s effectiveness which could also be considered as « prospective » : although analyses were retrospective and CATIs were not randomized, both the exposition (CATIs) and outcome (cholera suspected cases) were routinely and prospectively recorded.

- Discussion: « … selection of interventions did not appear to be based on evidence of their effectiveness. »

PLOS authors have the option to publish the peer review history of their article (what does this mean?). If published, this will include your full peer review and any attached files.

Reviewer #1: Yes: Godfrey Bwire, MBCHB, MPH, PhD

Reviewer #2: No
---

## [Decision Letter · Decision Letter 1]

8 Nov 2021

Dear Dr. Spiegel,

Thank you very much for submitting your manuscript "Case-area targeted preventive interventions to interrupt cholera transmission: current implementation practices and lessons learned" for consideration at PLOS Neglected Tropical Diseases. As with all papers reviewed by the journal, your manuscript was reviewed by members of the editorial board and by several independent reviewers. In light of the reviews (below this email), we would like to invite the resubmission of a significantly-revised version that takes into account the reviewers' comments. 

We cannot make any decision about publication until we have seen the revised manuscript and your response to the reviewers' comments. Your revised manuscript is also likely to be sent to reviewers for further evaluation.

Sincerely,

Jade Benjamin-Chung

Guest Editor

Emily Gurley

Deputy Editor

Reviewer's Responses to Questions

**Key Review Criteria Required for Acceptance?**

**Methods**

-Are the objectives of the study clearly articulated with a clear testable hypothesis stated?

-Is the study design appropriate to address the stated objectives?

-Is the population clearly described and appropriate for the hypothesis being tested?

-Is the sample size sufficient to ensure adequate power to address the hypothesis being tested?

-Were correct statistical analysis used to support conclusions?

-Are there concerns about ethical or regulatory requirements being met?

Reviewer #1: Are the objectives of the study clearly articulated with a clear testable hypothesis stated? YES

-Is the study design appropriate to address the stated objectives? YES

-Is the population clearly described and appropriate for the hypothesis being tested? NO

-Is the sample size sufficient to ensure adequate power to address the hypothesis being tested? YES

-Were correct statistical analysis used to support conclusions? YES

-Are there concerns about ethical or regulatory requirements being met? NA

Reviewer #2: (No Response)

**Results**

-Does the analysis presented match the analysis plan?

-Are the results clearly and completely presented?

-Are the figures (Tables, Images) of sufficient quality for clarity?

Reviewer #1: Does the analysis presented match the analysis plan? YES

-Are the results clearly and completely presented? YES

-Are the figures (Tables, Images) of sufficient quality for clarity? YES

Reviewer #2: (No Response)

**Conclusions**

-Are the conclusions supported by the data presented?

-Are the limitations of analysis clearly described?

-Do the authors discuss how these data can be helpful to advance our understanding of the topic under study?

-Is public health relevance addressed?

Reviewer #1: Are the conclusions supported by the data presented? NO. There is selective use of data

-Are the limitations of analysis clearly described? NO

-Do the authors discuss how these data can be helpful to advance our understanding of the topic under study? YES

-Is public health relevance addressed? YES

Reviewer #2: (No Response)

**Editorial and Data Presentation Modifications?**

Reviewer #1: Thank you very much for informing the authors to include the line numberiung. This version was userfrendly to review.

Reviewer #2: (No Response)

**Summary and General Comments**

Reviewer #1: Reviewer’s report

 Reviewer: 1

Manuscript No: PNTD-D-21-00779 

Manuscript title: Case-area targeted preventive interventions to interrupt cholera transmission: current implementation practices and lessons learned

 General comments

The authors have made efforts to address the issues raised with the original manuscript. This revised version is clearer and the study undoubtedly has the potential to streamline and strengthen CATI implementation due to the good recommendations listed by the authors. I applaud the authors for these well thought recommendations. However, in this updated version there are still important issues in the method section that need to be clarified to increase readability and allow for replicability of the study findings by other researchers.

Essential comments 

1. Method section. Lines 105-107, “The search was limited to publications between January 2009 and November 2019; English language publications were included in both searches, in addition to French and Spanish publications in the grey literature search.” and lines 96-97, “ A mixed-methods approach to study CATI implementation was employed, including: 1) reviews of peer reviewed journal publications and grey literature published in the past ten years; . 

The authors state that the study was limited to ten years however this is approximately 11 years. Furthermore, lines 164, table 1, the authors included studies in Cameroon, Duala, 2004 Reference 11 and in Kenya, Nyanza, 2008, Reference 12. This information that is not clear on the period of literature included makes it difficult for replicability of this study findings by other researchers. Therefore, the authors should revise the paper and clear inconstancies/ inaccuracies so as to make this study easily replicable by other researchers. The studies in Kenya and Cameron that are clearly out the period stated should be omitted. 

2. Abstract and method sections. Lines 40-42, “ Conclusions/Significance: CATIs are believed to be effective in reducing cholera outbreaks, but there is limited and context specific evidence of their effectiveness in reducing the incidence of cholera cases and lack of guidance for their consistent implementation.”. in this statement, the author note that there is limited evidence on effectiveness of CATI yet there are other studies such that by Bompangue et al 2020, https://doi.org/10.1186/s12879-020-4916-0. are not included. Yet, lines 129-131, “Retrospective case studies were used to investigate CATI implementation in DRC (2017-2020), Haiti (2010-2019), Yemen (2016-2020), and Zimbabwe (2018-2019) where the approach was implemented to control cholera outbreaks. Locations were selected in …” included the period 2020.. Therefore to avoid misinterpretation of the selective use of literature and information, the authors should revise this manuscript and include this study or in their discussion should refer to it as new finding that have weakened/overshadowed their study findings. 

3. Abstract. Lines 40-42, “ Conclusions/Significance: CATIs are believed to be effective in reducing cholera outbreaks, but there is limited and context specific evidence of their effectiveness in reducing the incidence of cholera cases and lack of guidance for their consistent implementation.”. my main concern with this conclusion is that the authors use the term “believed” where there is clear and robust study conducted by a competent team in a place (Bangladesh ) that has shapped the current knowledge on the epidemiology of cholera (Sack et al, https://doi.org/10.1093/infdis/jiab440). Therefore, the authors should revise the statement and remove the word believe since the facts are available. This revision should be carried out in the entire manuscript where the term believed is used.

4. Abstract and method sections. Abstract Lines 29-32, “ We identified 11 peer-reviewed and eight grey literature articles documenting CATIs and completed 30 key informant interviews in case studies in Democratic Republic of Congo, Haiti, Yemen, and Zimbabwe. We documented 15 outbreaks in countries where CATIs were used.”. and method section lines 128-135. All 100% of the countries listed in this statement are fragile states. Further, majority of the studies shaping the finding were from the fragile states (https://thedocs.worldbank.org/en/doc/179011582771134576-0090022020/original/FCSFY20.pdf ). Fragility is known to affect social services even when effective approaches are applied ( doi: 10.1093/heapol/czz142; https://www.usip.org/sites/default/files/resources/SR_301.pdf: http://www.gsdrc.org/docs/open/con86.pdf ). When the study in Kenya (2008) is excluded, this effect becomes even more clearer. Therefore, could it be that the observed results are due to fragile nature of the states where studies were carried out? The authors will need to explain in the discussion section the effect of this on their findings. 

Other comments

1. Lines 96-99, 

“A mixed-methods approach to study CATI implementation was employed, including: 1) reviews of peer reviewed journal publications and grey literature published in the past ten years; and 2) four retrospective case studies of recent cholera outbreaks in the Democratic Republic of the Congo (DRC), Haiti, Yemen, and Zimbabwe” 

and lines 29-32,

 “ Methodology/Principle Findings: We investigated implementation of cholera case-area targetedinterventions (CATIs) using systematic reviews and case studies. We identified 11 peer-reviewed and eight grey literature articles documenting CATIs and completed 30 key informant interviews in case studies in Democratic Republic of Congo, Haiti, Yemen, and Zimbabwe. We documented 15 outbreaks in countries where CATIs were used” 

The authors use the term “recent” that is open to misinterpretation by readers and scientists interested in replicability of the study findings. Therefore, the authors should clearly specify the timeframe/period to allow for replicability of the findings. 

2. In reference my earlier comment (first comment) on the original version which is not fully addressed yet. The authors interpreted rapid response team as CATI, yet this is WHO-AFRO strategy for outbreak investigation and response that is applicable to any infectious disease epidemic in WHO African region. Since most the studies are from WHO Afro region, I think that it would be important if the authors could raise this as a limitation in the interpretation of the study findings and conclusion.

END

Reviewer #2: * I sincerely acknowledge authors for their modifications and clarifications, and I believe I am now in line with the authors’ point of view.

* In the conclusion of their abstract, I however think the verb « believe » wrongly suggests that CATI effectiveness is a matter of unscientific belief. I suggest a more neutral sentence, for instance like this : « CATIs appear effective in reducing cholera outbreaks, but the available evidence remains limited and context specific, and no guidance for their consistent implementation has been released so far. »

* I understand the fact that the grey publication search was limited to the period January 2009 - November 2019. Considering the limited number of scientific publication on the subject, I really think that this period should be extended for them. In particular, the paper from Bompangue et al 2020 should really be cited in a systematic review published in 2022 and with a strong expected impact.

* I don’t understand the position of the new ref [5] (Voelkel 1971). Although it rapidly lists cholera transmission pathways, this French pioneering paper mainly describes principles of CATIs.

* I still strongly think that authors should also include the Sierra Leone and Guinea in 2012 report in their systematic review, even though it does not significantly change authors conclusions. This report provides valuable data on CATI implementation and impact, and to be systematic, this review should definitely cite this report.

* Concerning Comoros, a paper was published in 2007 describing response teams (https://www.researchgate.net/publication/229983054_Needs_for_an_Integrative_Approach_of_Epidemics_The_Example_of_Cholera/link/59e5d180a6fdcc1b1d96f21d/download), and I believe it should be included in this systematic review as well.

* Page 14: As far as I know, no reactive OCV was implemented in Haiti; and no vaccinator was member of CATI teams in Haiti. As the sentence mix Haiti and South Sudan, it is confusing.

* Concerning RDT and culture use for CATI implementation in Haiti, authors could easily complete their case study with informants already interviewed from Unicef, University of Notre Dame, or Assistance Publique Hôpitaux de Marseille.

* I am sorry I do not understand the source of the $1,776 monthly cost in Yemen, as the Unicef reference [13] states : « In Yemen, the average monthly cost range is US$1,500,000 – 1,875,000 for an average of 625 teams, with costs varying depending on rural and urban settings.43 This results in an average monthly cost of approximately US$2,400 for urban teams and US$3,000 for rural teams. » (p. 22)

* Line 284-285: it is unclear to me what « antibody prophylaxis » state for. It usually refers to the use of monoclonal antibodies for post-exposure prophylaxis against an infectious diseases (such as rabies, hepatitis B, or COVID-19). Of course, this does not exist for cholera. I think authors aimed to refer to « antibiotic prophylaxis ».

PLOS authors have the option to publish the peer review history of their article (what does this mean?). If published, this will include your full peer review and any attached files.

Reviewer #1: Yes: Godfrey Bwire, MBChB, MPH, PhD

Reviewer #2: No
---

## [Editor Report · Decision Letter 2]

1 Dec 2021

Dear Dr. Spiegel,

We are pleased to inform you that your manuscript 'Case-area targeted preventive interventions to interrupt cholera transmission: current implementation practices and lessons learned' has been provisionally accepted for publication in PLOS Neglected Tropical Diseases.

Best regards,

Jade Benjamin-Chung

Guest Editor

Emily Gurley

Deputy Editor

---

## [Editor Report · Acceptance letter]

13 Dec 2021

Dear Dr. Spiegel,

We are delighted to inform you that your manuscript, "Case-area targeted preventive interventions to interrupt cholera transmission: current implementation practices and lessons learned," has been formally accepted for publication in PLOS Neglected Tropical Diseases.

Best regards,

Shaden Kamhawi

co-Editor-in-Chief

Paul Brindley

co-Editor-in-Chief
